# Investigation of Consequences of High-Voltage Pulsed Electric Field and TGase Cross-Linking on the Physicochemical and Rheological Properties of *Pleurotus eryngii* Protein

**DOI:** 10.3390/foods12030647

**Published:** 2023-02-02

**Authors:** Jiaxin Li, Yan Feng, Qianying Cheng, Jingyu Liu, Shaojun Yun, Yanfen Cheng, Feier Cheng, Jinling Cao, Cuiping Feng

**Affiliations:** 1College of Food Science and Engineering, Shanxi Agricultural University, Taigu, Jinzhong 030801, China; 2Department of Life Science, Lyuliang University, Lyuliang 033001, China; 3Shanxi Key Laboratory of Edible Fungi for Loess Plateau, Taigu, Jinzhong 030801, China

**Keywords:** *Pleurotus eryngii* protein, high-voltage pulsed electric field, TGase, physicochemical properties, rheological properties

## Abstract

This study aimed to evaluate the effects of high-voltage pulsed electric fields (HPEF) and transglutaminase (TGase) cross-clinking on the physicochemical and rheological properties of *Pleurotus eryngii* protein (PEP). The results showed that HPEF increased α-helixes and β-turns but decreased β-folds. A HPEF at 1500 V/cm maximized the free sulfhydryl content and solubility of PEP. TGase formed high-molecular-weight polymers in PEP. TGase at 0.25% maximized the free sulfhydryl groups, particle size, and solubility; shifted the maximum absorption wavelength from 343 nm to 339 nm and 341 nm; increased α-helixes and β-turns and decreased β-folds; and showed better rheological properties. Compared with TGase cross-linking, HPEF-1500 V/cm and 1% TGase significantly reduced the free sulfhydryl groups, particle size, and solubility, produced more uniform network structures, and improved the rheological properties. These results suggest that HPEF can increase the cross-linking of TGase and improve rheological properties of TGase-cross-linked PEP by affecting the physicochemical properties.

## 1. Introduction

*Pleurotus eryngii* protein (PEP), extracted from *Pleurotus eryngii*, is a mixture of proteins including albumin, glutenin, globulin, and prolamin [1]. It has potential application value in functional foods based on its anti-inflammatory, antitumor, and immunoregulatory activities [2]. PEP has become a common ingredient in various food formulations owing to its high nutritional value and versatile functional properties, such as emulsification, gelation, foaming, and flavor-binding properties. These available properties of the protein can be tailored or improved by modification for enhancing its functional qualities and nutritional potential [3]. Several methods have been developed to improve the physicochemical and functional properties of proteins by changing the intermolecular aggregation, the forces maintaining the conformation of protein molecules and the advanced structure of proteins, such as moist heat, ultrasound, ultra-high pressure, glycosylation, deamidation, and enzyme catalysis [4,5]. In particular, enzymatic modification is a green and effective method due to its high specificity and efficiency and mild reaction conditions.

Transglutaminase (TGase), a kind of extracellular catalytic transferase, is commonly used for non-hydrolytic modification of bio-enzymes. It can catalyze inter- and intra-molecular glutamine (as acyl donor) and lysine (as acyl receptor) residues of proteins for acyl transfer reactions, deamidation reactions, and cross-linking polymerization reactions [6]. These reactions can improve the solubility, emulsification, foaming, gelation, viscosity, and water-holding capacity of proteins by altering their structure. However, albumin, glutenin, and globulin, as the main components of PEP, are compact globular proteins [7] and are thus less susceptible to cross-linking with TGase. Therefore, it can facilitate the cross-linking reactions to expose the enzyme-targeted sites [8]. Some experiments have been carried out to improve the degree of cross-linking [8,9].

High-voltage pulsed electric field (HPEF), a primary method of modifying proteins, is a non-thermal food processing and preservation method. It has the advantages of short processing time, low energy consumption, and environmental friendliness [10]. It changes the conformation of proteins by inducing the polarization of protein molecules and affecting the contents of hydrophobic groups, sulfhydryl groups, disulfide bonds, etc., thus causing protein denaturation and aggregation [11]. In recent years, by optimizing PEF parameters, some plant proteins (such as canola protein and gluten protein concentrate) have been efficiently modified with improved solubility and other functional properties [12,13]. However, the modification effect seems to be unsatisfactory. For example, PEF treatment does not significantly increase the solubility of gluten protein concentrate and the improve water-holding capacity and oil-holding capacity. This level of protein modification is obviously insufficient to meet the needs of the food industry. Thus, it is urgent to inquire into other techniques to further improve the efficiency of HPEF-induced protein modification.

Therefore, to better improve the structure and functional properties of PEP, complex modification of HPEF and TGase was used to explore the effects of HPEF and TGase on the relationship between microstructure and functional properties of PEP, such as the secondary structure, molecular weight, free sulfhydryl groups, particle size distribution, and apparent morphology, as well as the rheological properties of the protein. This study can be helpful to better understand the development of HPEF and TGase in improving PEP and provide a reference for improving the comprehensive utilization of *Pleurotus eryngii*.

## 2. Materials and Methods

### 2.1. Materials

Five kilograms of *Pleurotus eryngii* was provided by China Taigu Edible Fungi Engineering and Technology Center (Jinzhong, Shanxi, China) and was divided into three batches for subsequent experiments. The following materials were also used: TGase (TG-B, food grade, 120 U/g, Jiangsu Yiming Biological Co., Ltd., Taixing, China), Bovine Serum Albumin (A391210g recover Sigma, St. Louis, MO, USA), Gmur 250 Coomassie brilliant Blue (Shanghai Yuchuang Biotechnology Co., Ltd., Shanghai, China), 5,5′-dithiobis-(2-nitrobenzoic acid) (DTNB) (Wuxi Tongchuang Technology Co., Ltd., Wuxi, China), ethylenediamine tetraacetic acid (EDTA) (Changzhou Dehao Chemical Co., Ltd., Changzhou, China).

### 2.2. Preparation of PEP

*Pleurotus eryngii* was dried and crushed to 150 mesh with an ultra-fine grinding vibration mill (WFM-10, Jiangyin Xiangda Machinery Manufacturing Co., Ltd., Jiangyin, China) to prepare protein for the follow-up experiment. PEP was extracted using the Osborne graded extraction method [14]. Briefly, *Pleurotus eryngii* powder was extracted with distilled water, NaCl (3% *w*/*v*), and NaOH (0.01% *w*/*v*, pH 12) for 2.5 h, 2 h, and 3 h in order to obtain albumin, globulin, and glutenin at isoelectric points of 3.9, 4.2 and 4.3, respectively. All the solid:liquid ratios used in the extractions were 1:12. The obtained proteins were mixed and dialyzed at 4 °C for 12 h and then were subjected to liquid nitrogen and free-dried with a vacuum freeze-dryer (SHIA-10A-50A, CHRIST, Germany) at −30 °C for 6 h and 20 °C for 48 h. Then the PEP was stored at −18 °C. The experiment was repeated three times to determine that 100 g of *Pleurotus eryngii* powder could obtain about 13 g PEP.

### 2.3. Treatment of PEP

#### 2.3.1. Treatment of TGase

The PEP was dispersed in deionized water (containing 10% protein, *w*/*w*). Next TGase was added to the PEP solution with final concentrations of 0.25%, 0.5%, 1%, 2%, and 4% (*w*/*w*, PEP base). The mixture was then shaken well and reacted in a thermostatic water bath (DK-S10, Shanghai Boxun Industrial Co., Ltd., Shanghai, China) at 50 °C for 4 h, without avoiding light during the whole reaction. Afterwards, the reaction was stopped by inactivating the enzyme at 75 °C for 15 min. Finally, liquid nitrogen and vacuum freeze-drying were used to obtain TGase cross-linked PEP with a 92% final yield of the product. The test was repeated three times.

#### 2.3.2. Treatment of HPEF

The HPEF equipment includes a pulsed electric field generator (ECM830) and an electrode treatment chamber consisting of two 20 × 20 mm stainless steel plates with adjustable spacing (BTX company, Holliston, MA, USA). The optimal conditions were obtained with the HPEF treatment pre-test, and the mode was adjusted to high voltage (HV) mode. Different intensities of HPEF (E = 500, 1000, 1500, 2000, 2500 V/cm) were respectively applied with the fixed pulse width τ = 45 μs and uniform pulse number n = 90 to obtain the HPEF-treated PEP.

#### 2.3.3. Combined Treatment of HPEF and TGase

The HPEF-treated PEP was dispersed in deionized water (containing 10% protein, *w*/*w*). Then 1% TGase (*w*/*w*, PEP base) was added separately, shaken well, and reacted in a 50 °C water bath for 4 h, after which the reaction was stopped by inactivating the enzyme at 75 °C for 15 min. Finally, the HPEF-TG-PEP was obtained by treatment with liquid nitrogen and vacuum freeze-drying.

### 2.4. Physicochemical Properties 

#### 2.4.1. Free Sulfhydryl Group

The PEP and differently treated PEP samples were diluted to 2 mg/mL with a reaction buffer containing 0.1 M sodium phosphate buffer (PBS, pH 7.4) and 1 mM EDTA (pH 8.0), stirred at room temperature for 60 min, and centrifuged at 10,000 r/min for 15 min. Then 0.5 mL supernatant and 0.1 M Ellman’s reagent solution were added to 5 mL of the above reaction buffer at the same time. After reaction in the dark for 15 min, the absorbance was measured at 412 nm using a UV-Vis spectrophotometer (UV9100A, Beijing LabTech Co., Ltd., Beijing, China) with the above reaction buffer as blank. The experiment was repeated three times. The concentration of protein was calculated according to the correction curve of cysteine (0.25–1.5 mg/mL). The equation for the correction curve of cysteine is as follows:A_412_ = 1.1647 × C + 0.0327

In the formula, A represents the absorbance and C represents the protein concentration.

#### 2.4.2. Scanning Electron Microscopy (SEM) Analysis

The dried samples were mounted on a copper stake with a double-sided carbon tag and coated with a layer of platinum. Then the microscopic morphology was observed with a JSM-6490LV SEM (JEOL Electronics Co., Ltd., Tokyo, Japan) and taken using the accompanying software at an accelerating voltage of 15 kV.

#### 2.4.3. SDS-PAGE

According to Laemmli’s SDS-PAGE method [15], the protein samples dissolved in the sample buffer were oscillated completely and then fully reacted in boiling water for 5 min. After being cooled, the samples were centrifuged to obtain the supernatant. 

The electrophoresis gel was prepared with 12% separation gel and 5% concentrated gel. A 5 μL Color Prestained Protein Maker (10~180 kDa, Beijing Bioss Biological Co., Ltd., Beijing, China) and 20 μL protein were added to the electrophoresis bath (Mini-PROTEAN, Bio-rad Co., Hercules, CA, USA). The voltage of concentrated gel and separation gel were, respectively, 100 V and 150 V. After being stopped at a distance from the bottom 1 cm, the electrophoretic gel was dyed, decolorized, and then scanned for graphical analysis.

#### 2.4.4. Particle Size and Zeta-Potential

The protein samples were diluted with 0.1 M PBS (pH 7.4) to 1 mg/mL for three times at 25 °C. The particle sizes and Zeta-potential of PEP samples were respectively determined with a particle size analyzer (HL2020-B, Beijing haixinrui Technology Co., Ltd., Beijing, China) and a potential analyzer (Zeta Plus, Brookhaven Co., Ltd., Holtsville, NY, USA) [16].

#### 2.4.5. Intrinsic Fluorescence Emission Spectrum

The protein samples were diluted to 1 mg/mL with PBS (pH 7.4). The intrinsic fluorescence emission spectra were measured using a Hitachi F4500 fluorometer (Hitachi Co., Ltd., Tokyo, Japan) with an excitation wavelength of 280 nm, a slit width of 5 nm, and an scanning wavelength of 290~430 nm [17].

#### 2.4.6. Fourier Transform Infrared (FTIR) Spectroscopy

The dry sample of 1 mg and potassium bromide of 100 mg were mixed in the mixing mortar to scan the FTIR spectra using an FTIR spectrometer (Madison Nicolet Is 10, Thermo Nicolet Co., Waltham, MA, USA) at a full wavelength (4000–400 cm^−1^). The spectrum was recorded at 32 scans with a resolution of 4 cm^−1^. According to a previous method [18], the PeakFit v4.04 software was used for baseline correction, Gaussian deconvolution, and second derivative fitting. The assignment of each peak was determined, and the changes in secondary structural elements (α-helix, β-sheet and β-turn) were quantified in untreated and treated PEP samples.

### 2.5. Rheological Property

#### 2.5.1. Steady Rheological Properties of PEP Solution

Samples were obtained by dissolving 8 g of PEP powder in 100 mL of distilled water and analyzed with a MCR-102 rheometer from Antonpa Company (Grza, Austria) [19]. The experimental conditions were 25 ± 0.1 °C, 50 mm, 1° lamina, 0.6 mL PEP solution, 0.103 mm interlaminar space, and scraping off the excess protein solution. In the linear scanning mode, a steady shear rate varying from 0 to 100 s was used to measure the shear stress variation with the shear rate. The steady rheological fitting curve of the PEP protein solution was fitted linearly using the Herschel–Bulkley model τ = τ HB + c γ P, where τ HB represents the yield stress, c is the viscosity coefficient, P is the flow characteristic index, and γ is the shear rate [19]. 

#### 2.5.2. Thixotropy of PEP Solution

The thixotropy of the 3% PEP solution was measured using an MCR-102 Antonpa rotary rheometer in two-step steady shear mode, and the data were collected with RheoCompass^TM^ software 1.31.43 (Antonpa Co., Ltd., Graz, Steiermark, Austria). The test conditions were 25 ± 0.1 °C, 50 mm, 1° lamina, zero gap of 0.103 mm, and the excess protein solution was scraped off; the shear rate increased from 0 s^−1^ to 130 s^−1^ and then decreased from 130 s^−1^ to 0 s^−1^.

### 2.6. Solubility

The PEP samples were diluted to 5 mg/mL with distilled water and centrifuged at 10,000 r/min for 15 min. The protein content of the supernatant was determined with the colorimetric method using Kemas Brilliant Blue reagent (0.01% Kemas Brilliant Blue G-250, 4.7% ethanol, 8.5% H_3_PO_4_) at 595 nm [20]. The protein concentration was calculated based on the calibration curve of bovine serum protein (0.2–1 mg/mL) using the following equation.
A_595_ = 0.5451 × C + 0.0173
Solubility (%) = (Supernatant protein mass/Total protein mass) × 100%

In the formula, C is the mass concentration of the protein, R^2^ = 0.9997.

### 2.7. Statistical Analysis

Results are shown as mean ± standard deviation (n = 3), and analysis of variance (ANOVA) was expressed using SPSS System Software 22.0 (IBM Co., Armonk, NY, USA). Before analysis, Kolmogor-ov–Smirnov one-sample test and Levene’s test were used to determine the normal distribution and homogeneity of variance, respectively. No violation of the assumptions for ANOVA was detected. Significant differences between the individuals were tested with Duncan’s multiple ranges (*p* < 0.05). 

## 3. Results

### 3.1. Free Sulfhydryl Group

As shown in Figure 1, the free sulfhydryl group of PEP increased and then decreased as the field strength of HPEF increased, with the highest free sulfhydryl group content of 0.188 mM at 1500 V/cm HPEF. The exposure of free sulfhydryl groups may be caused by the unfolding or the ionization of free sulfhydryl groups in proteins. A similar result was found in HPEF-treated ovalbumin [21]. Moreover, HPEF also significantly increases the free sulfhydryl groups of globulins, albumins, and whole proteins in rapeseed [13].

As shown in Figure 1, TGase (0.25%) significantly increased the free sulfhydryl content of PEP by unfolding the structure of PEP to form an unfolded molecular structure and expose the internal sulfhydryl groups. A similar result was obtained in the free sulfhydryl content of peanut isolate [22]. Moreover, the sulfhydryl content was increased within a range of TGase dosages. However, the free sulfhydryl content of PEP was significantly reduced with TGase addition (0.5–2%), supported by the studies on soybean protein and rice flour [23,24]. The decrease in the free sulfhydryl groups of the protein may be because the exposed sulfhydryl groups are easily oxidized to form disulfide bonds, which in turn leads to a decrease in their content [25]. The extensive polymerization of protein through disulfide bonds after cross-linking of TGase may explain the reduction in free sulfhydryl content [26].

As shown in Figure 1, the compound treatment caused the free sulfhydryl content of PEP to decrease and then increase, with the lowest free sulfhydryl content after the compound treatment with 1500 V/cm HPEF and 1% TG. This indicates that the compound treatment may slightly damage the structure of PEP, making the TGase cross-linking reaction more complete and encapsulating the free sulfhydryl groups in the structure.

### 3.2. SEM

The HPEF treatment thickened the compact lamellar structure of PEP and produced more fragments, which made the cut surface more irregular compared with the untreated group. These changes may be owing to the electric field, which expends PEP molecules and makes them more fluffy, with the most significant change at an HPEF intensity of 2500 V/cm (Figure 2a–f).

The control PEP showed a thin sheet structure (Figure 2g), while the TGase-cross-linked PEP showed a network structure (Figure 2h–l), which indicates that TGase catalyzes the formation of covalent cross-linking between protein molecules to form an intramolecular or intermolecular network structure. When 0.25% TGase was added, a small network structure appeared in PEP; the molecular network structure gradually increased, and a more uniform and porous network structure appeared in PEP treated with 2% TGase. A similar result was found in amygdalin gels [27]. When 4% TGase was added, the PEP hydrogel structure showed disorganization and collapse, which could be attributed to excessive cross-linking of TGase, supported by a previous study on soy protein [28]. In addition, the increase in cross-linked aggregates may correlate with the decrease of PEP solubility after TGase cross-linking. A similar result was obtained in a study of whey–soybean mixed protein [29]. Figure 2n–r shows the PEP after the composite treatment. The aggregates in Figure 2o were denser, which indicates that the TGase cross-linking reaction is effectively combined under HPEF (1500 V/cm) treatment [30].

### 3.3. SDS-PAGE

SDS-PAGE was used to detect the changes in molecular weight of PEP after different treatments. The electrophoretic bands above 180 kDa in the controls were observed, unlike in whey protein [31] and soy protein [32]. The intensity of PEP subunit bands after HPEF treatment was similar to control PEP (Figure 3b), indicating that HPEF treatment does not cause dissociation of PEP and generate new subunit bands, coincident with Qian [18], who investigated PEF on egg white protein.

The band distribution of untreated PEP and TGase-treated PEP was similar, but the color of electrophoretic bands below 180 kDa gradually became lighter or disappeared with the increase in TGase dosage (Figure 3a). Moreover, after TG cross-linking, new bands were generated at the inlet, which indicates that the TGase-treated PEP forms high-molecular-weight aggregates above 180 kDa, and the cross-linked aggregates cannot pass on the concentrated and separated gels. In addition, the PEP formed blocky colloidal material after TGase cross-linking pretreatment, which was supported by a previous study [33]. The formation of blocky colloidal material may be caused by acyl transfer reactions between the glutamyl bonds and lysine side chains of proteins promoted by TGase resulting in the formation of more soluble aggregates and polymers [34]. In addition, TGase-induced aggregate formation also suggests that PEP has substrate efficacy for TGase [27].

Compared with PEP cross-linked by TGase alone, complex treatment (HPEF-1500 V/cm; 1% TGase) produced lighter electrophoretic bands and deeper bands above 180 kDa at the inlet (Figure 3b), which indicates that the treatment with 1500 V/cm HPEF may expose some active functional groups previously buried inside the protein molecules and change the protein conformation, thus promoting the TGase cross-linking reaction and resulting in a larger molecular weight formed by cross-linking at the same enzyme amount.

### 3.4. Particle Size Distribution, Average Particle Size

The particle size distribution curves of PEP after HPEF treatment were to the left of untreated PEP (Figure 4a), which indicates that HPEF treatment breaks the structure of PEP, making the average particle size of PEP smaller and generating more binding sites for TGase and PEP. Combined with the average particle size after HPEF treatment (Figure 4d), the average particle size of the protein tended to decrease first and then increased, and its average particle size was 153.9 nm at 2000 V/cm of HPEF. The same result was obtained in a study of soybean protein isolates, which may be due to the dissociation and reaggregation of proteins induced by PEF treatment [35].

The particle size distribution of PEP cross-linked by TGase after incubating PEP and TGase at 50 °C for 4 h is shown in Figure 4b. An overall rightward shift of the PEP curve after TGase (0.25%) cross-linking can be found in Figure 4b, which indicates that the right amount of TGase cross-linking can form soluble aggregates and increase the average particle size of PEP in a moderate amount. A similar result was found in a previous study [22]. With the increase in TGase addition (0.5–4%), the particle size distribution curve of PEP soluble protein kept shifting to the left, and the average particle size kept decreasing significantly, which shows that a trace amount of TGase increases the particle size of the soluble protein. The more TGase was added, the more insoluble aggregates of PEP were formed, which reduce the soluble protein particle size and increase the insoluble protein particle size. SDS-PAGE also proved the reliability of the results.

The particle size distribution curves of the composite-treated PEP were all on the left side of the untreated PEP (Figure 4c), which indicates that the PEP may have formed insoluble protein aggregates. The average particle size of PEP after composite treatment (HPEF-1500, 2000 V/cm; 1% TGase) was significantly smaller than that of PEP treated with 1% TGase alone, which indicates that the cross-linking reaction between PEP and TGase is promoted after 1500 V/cm HPEF pretreatment. Similar to the present result, TGase cross-linking increased the mean particle size of whey proteins, which when cross-linked with TGase after sonication pretreatment was larger than that of whey proteins cross-linked with TGase [33].

### 3.5. Zeta-Potential

The value of the Zeta-potential can be significantly correlated with the stability of a colloidal dispersion and indicates the degree of repulsion between neighboring substances and similarly charged particles [36]. For sufficiently small molecules and particles, a higher Zeta-potential will bring stability. When the potential is low, the attraction is greater than the repulsion, and the dispersion will break up and flocculate [37]. An absolute value of Zeta-potential less than 30 mV indicates that the suspension is unstable and, conversely, a total value of Zeta-potential greater than 30 mV indicates that the suspension system is stable. As shown in Figure 4e, the absolute values of potential values in all groups exceeded 30 mV, but there were no significant differences between the control and treated groups, which indicates that different treatments do not affect the stability of protein solutions. Similarly, Zhang et al. [33] found that the TGase cross-linking reaction had no significant impact on the Zeta-potential of whey protein soluble aggregates.

### 3.6. Intrinsic Fluorescence Emission Spectrum

In natural proteins, the residues that can fluoresce are tryptophan, tyrosine, and phenylalanine. Due to the difference in aromatic groups on the side chains, these three amino acid residues have different fluorescence spectra with maximum absorption wavelengths of 348, 303, and 282 nm, respectively, which depends mainly on the contribution of tryptophan residues [38]. According to the aromatic properties, tryptophan is completely or partially buried in the hydrophobic core inside the protein or at the interface between two protein structural or substructural domains [39]. Meanwhile, the fluorescence quantum yield of tryptophan can identify changes in protein structure [40].

As shown in Figure 5a, compared with the λ_max_ (343 nm) in the untreated group, the maximum absorption wavelength after HPEF treatment was less than 343 nm with no fixed pattern, which indicates that HPEF treatment may have caused a slight blue shift. Meanwhile, the fluorescence intensity tended to increase and then decrease with the rise in HPEF intensity. The fluorescence intensity increased slightly at the HPEF intensity of 500 V/cm and decreased to the lowest at the HPEF intensity of 2500 V/cm. The changes in fluorescence intensity after HPEF treatment indicate that HPEF alters the spatial structure of proteins and destroys the hydrophobic groups inside the protein molecules at the weaker electric field intensity, thus exposing more hydrophobic groups outside the protein molecules and increasing the fluorescence intensity. It has also been suggested that HPEF treatment will expose more previously buried tryptophan residues [18]. Moreover, the change in HPEF may lead to fluorescence quenching of tryptophan residues, but the effect of HPEF on them remains to be further studied.

As shown in Figure 5b, the λ_max_ was 339 nm for 0.25–2% TGase and 341 nm for 4% TGase, which indicates that an obvious blue shift is produced after TG cross-linking. Moreover, the intensity of PEP treated with TGase (0.25%) was higher than that of the control PEP, which reflects that the TGase cross-linking reaction may lead to the exposure of buried tryptophan residues in the PEP. This is consistent with previous reports [9,17]. This can be explained by the fact that some tryptophan residues may bind to non-polar regions due to aggregation or peptide–peptide binding, resulting in an increase in relative fluorescence intensity. In addition, the fluorescence intensity of PEP decreased with increasing amounts of TGase (0.5–4%), suggesting that the TGase-induced polymerization masking effect may have caused the chromophores (tryptophan, tyrosine, and phenylalanine residues) to be masked, supported by a previous study [41].

Compared with the 1% TGase group, the λ_max_ of the composite treatment groups did not produce a blue-shift or red-shift phenomenon (Figure 5c). With the increasing HPEF intensity in the combined treatment groups, the fluorescence intensity of PEP decreased and then increased and was the lowest after treatment with HPEF-1000 or HPEF-1500 V/cm and 1% TGase. The decrease in fluorescence intensity after HPEF treatment may be attributed to the fact that HPEF under the proper conditions will disrupt the spatial protein structure, expose more TGase binding sites, and bury the chromogenic groups such as tryptophan into the protein molecules.

### 3.7. Secondary Structure

The subunit composition of PEP was significantly changed by SDS-PAGE, from which it could be inferred that its secondary structure was also changed to some extent. To clearly understand the stretching and aggregation properties of the protein, the changes in the secondary structure of PEP before and after treatment were investigated by analyzing the planar bending of NH, the change in -CN length in the amide one band (1700–1600 cm^−1^) region of the IR spectrum, and the change in C=O length in the structure [22,42].

As shown in Table 1, HPEF increased α-helixes and β-turns but decreased β-folds of PEP. With the growth of the HPEF strength, the α-helixes and β-folds gradually reduced, and the β-turns continuously increased. Moreover, the overall decline in β-folds was equal to the increase in α-helixes and β-turns, which indicates that HPEF might make the PEP structure less rigid and more flexible. A similar result was obtained in a previous study of rapeseed globulin [13].

After PEP was cross-linked with TGase, the α-helix and β-turn content increased and the β-fold content declined, which indicates that TGase cross-linking may promote the transformation of β-folds to α-helixes and β-turns. The presence of more hydrogen bonds in α-helixes and β-folds makes the protein structure exhibit a certain rigidity, while β-turns and irregular curls show great flexibility because they do not contain hydrogen bonds and other interactions [43]. A study by Marcoa and Rosell [44] also reported that large polymers and aggregates formed with extensive TGase cross-linking reactions might reduce the flexibility of the protein. The change in the secondary structure content of PEP after modification shows that the increase in α-helixes is significantly less than the decrease in β-folds, so TGase may make the PEP structure less rigid and more flexible. However, TGase cross-linking induced a decline in α-helixes and β-turns and an increase in β-folds compared to untreated PPI [22], contrary to our study, which may be due to the different structures of the substrate proteins cross-linked with TGase, resulting in inconsistent effects on the secondary structure. In addition, the secondary structure content of compound-treated PEP did not differ significantly, suggesting that HPEF treatment did not significantly synergize the TGase cross-linking effect and thus affect the secondary structure of PEP.

### 3.8. Rheological Properties

#### 3.8.1. Steady-State Rheology

##### Effects of HPEF on Steady Rheological Properties of PEP Solution

The shear stress (Pa) versus shear rate (s^−1^) was plotted for the control and HPEF-treated PEP to determine the flow behavior (N) of PEP (Figure 6a,b). Both untreated and treated PEP samples exhibited a non-Newtonian pseudoplastic fluid behavior (N < 1) and the shear stress of the samples increased significantly with the increasing shear rate. The shear stress of PEP decreased and then increased after low-strength HPEF and decreased significantly after HPEF (1000 V/cm) treatment, while it increased significantly after high-strength HPEF (2500 V/cm) treatment. These results confirm that HPEF has a greater effect on the shear stress of PEP, and high-strength HPEF treatment has a greater effect. A previous study on the rheological properties of soybean pulp supported our findings [45]. The apparent viscosity (Pa-S) of control and HPEF-treated samples was plotted against the shear rate (s^−1^) to determine the flow behavior of PEP (N) (Figure 6a,b). The apparent viscosity of PEP decreased significantly with the increasing shear rate, consistent with the power-law model of flow behavior without inter-particle interactions. These results were similar to those of soy protein emulsions under homogenous pressure treatment [46]. However, the protein structure collapsed rapidly at the initial shear and subsequently changed more slowly at higher shear rates, which can be attributed to the combined effect of the breakage of weak junctions between proteins and the reconstruction of such junctions as a result of Brownian motion and molecular collisions [47]. The viscosity coefficient (K) decreased continuously when the HPEF intensity increased from 500 V/cm to 1000 V/cm and from 1000 V/cm to 2500 V/cm (Table 2). The increase in apparent viscosity after HPEF treatment is thought to reflect intermolecular interactions due to the attraction between adjacent denaturing molecules, forming weak transient networks. These interactions can increase in effective volume. As protein aggregates become larger and occupy more space, they increase the apparent viscosity of the fluid system [48]. The increase in apparent viscosity of PEP treated with HPEF is similar to that of soy milk treated with PEP [46].

##### Effects of TGase on Steady-State Rheological Properties of PEP

Shear stress and apparent viscosity are important factors in determining the dilution of solutions. All samples exhibited a shear-thinning behavior in the shear rate range of 1–100 s^−1^. The shear stress increased and the apparent viscosity decreased with the increasing shear rate (Figure 6c,d). Therefore, all samples exhibited a pseudoplastic and non-Newtonian behavior in the shear rate range of 1 to 100 s^−1^ [49]. The shear stress and apparent viscosity of PEP increased rapidly with the increasing TGase concentration, which may result from protein molecular rearrangement in the cooling region. TGase reduces water mobility in the protein network and provides greater resistance to flow [34]. The apparent viscosity of PEP increased rapidly with the addition of TGase, suggesting that TGase promotes further cross-linking of protein molecular chains, resulting in higher apparent viscosity [50]. When 4% TGase was added, the shear stress increased about 10 times, and the apparent viscosity increased about 40 times compared to the untreated PEP, which was consistent with the solubility results. The higher solubility of PEP made the suspension flow more easily under shear [51]. As shown in Figure 6 and Table 3, TGase increased K but decreased the flow behavior index (N) of the TGase cross-linked PEP. The high-molecular-weight polymer formed by TGase cross-linking treatment may lead to higher apparent viscosity [26]. These results suggest that the increase in TGase doses promotes the degree of cross-linking of PEP. The elevated degree of cross-linking increases the consistency of the PEP solution and the K and decreases the N. Therefore, TGase cross-linking can promote protein aggregation during protein molecular chain rearrangement.

##### Effects of HPEF and TGase Treatment on Steady Rheological Properties of PEP Solution

Both control and composite-treated PEP samples exhibited a non-Newtonian pseudoplastic fluid behavior (N < 1) and the shear stress of the samples increased significantly with the increase in shear rate (Figure 6e,f, Table 4). Moreover, the shear stress of the composite-treated (HPEF-500, 1000 and 1500 V/cm; 1% TGase) PEP increased compared to the TGase-cross-linked PEP, while the shear stress of the composite-treated (HPEF-2000 and 2500 V/cm; 1% TGase) PEP decreased significantly compared to the TGase-cross-linked PEP. The low field strength of HPEF opened the structure of PEP to promote the cross-linking of TGase, which increases the PEP viscosity and the shear stress. In contrast, the high field strength of HPEF and TGase composite treatment made the PEP form insoluble aggregates, which made the solution viscosity smaller.

#### 3.8.2. Dynamic Rheological Properties of PEP Solution

##### Effects of HPEF on Dynamic Rheological Properties of PEP Solution

As shown in Figure 7a, in the frequency range of 1–100 rad/s, the storage modulus (G’) and loss modulus (G’) values of PEP showed irregular variation. The G’ and G” curves were not stable and parallel as the angular frequency increased. The G’ and G” curves of control and HPEF (500, 2000, 2500 V/cm)-treated PEP appeared to intersect, which was defined as the gel point temperature of the measured material. This point represents the transition stage from a substance considered a liquid to a more solid-like substance. The results of dynamic rheology show that the control and HPEF-treated solution systems are unstable, while the HPEF treatment does not make the PEP gel. Moreover, the tanδ values of PEP showed irregular changes with increasing angular frequency (Figure 7b), which also indicates that the HPEF has little effect on the dynamic rheological properties and does not promote the stability of the PEP solution system.

##### Effects of TGase on Dynamic Rheological Properties of PEP Solution

The G’ values were higher than the G” values in the frequency range of 1–100 rad/s (Figure 7c), indicating a dominant elastic behavior. Moreover, the dependence of these two moduli on the angular frequency was small, which means that the formed hydrogel is strong. The difference between the G’ and G” values of more than ten times also indicates that the hydrogel network is stable [52]. In addition, the control group had the lowest G’ and G” values, showing that PEP without the enzyme could not cross-link to form a gel. After the addition of TGase, gels are formed in PEP, which results in an increase in G’ and G” values. Moreover, the G’ and G” values increased with the rise in TGase dose, which indicates that both elastic and viscous moduli are increasing, probably due to the increase in protein–protein interactions and water–protein interactions under TGase activity [53]. G’ and G” values of gluten have been reported to increase with growing TGase levels, while tanδ values decrease with increasing TGase levels [24]. Furthermore, the tanδ value of PEP hydrogel keeps decreasing after TGase cross-linking (Figure 7d). The tanδ value of PEP in the control group was more significant than that at in angular frequency of 79~84 rad/s, and the tanδ value of PEP kept decreasing with the increase in TGase. The solution did not easily form a gel when the tanδ value > 1. The solution gel was more stable when the tanδ value was closer to 0.1, and the tanδ value < 0.1 was a strong gel. A previous study reported that the highest tanδ values of pea protein gels formed by thermal pretreatment and subsequent TGase cross-linking were less than 0.1 (recorded in the frequency ranges of 0.01 to 10 Hz), based on which the pea protein gels were further classified as strong gels [54].

##### Effects of HPEF and TGase Treatment on Dynamic Rheological Properties of PEP Solution

The G’ values were higher than the G” values in the frequency range of 1–100 rad/s (Figure 7e). As the field strength of HPEF increased from 500 V/cm to 1500 V/cm, the G’ value and G” value kept increasing, which indicates that PEP at low field strength will properly unfold the structure of PEP and expose more binding sites of TGase to promote the formation of a hydrogel. When the HPEF was increased from 2000 V/cm to 2500 V/cm, the G’ and G” values of PEP started to decrease, which indicates that the high field strength might excessively damage the structure of the protein or denature the PEP, resulting in the decrease in TGase cross-linking degree. In addition, the tanδ values of the composite treatment were lower than those of the PEP treated with TGase (Figure 7f), which indicates that the solution gelation is more stable and the combined treatment of HPEF and TGase is superior to the TGase treatment.

### 3.9. Solubility

The HPEF pretreatment (1000, 1500, 2000 V/cm) significantly increased the solubility of PEP compared to that of the control protein (35.76%). The increase in solubility is probably because the HPEF-induced unfolding or loosening of the protein molecules exposes the internal hydrophilic groups and promotes the air–water interface formation. A similar result was obtained in a previous study [13]. However, 2500 V/cm HPEF treatment exposed a higher number of hydrophobic groups, which in turn made PEP less soluble. The decreased solubility may be caused by the higher voltage, which produces more energy to be applied to the protein molecules. The interactions between the extended molecules constitute stable two-dimensional networks and interfacial membranes, thus affecting the solubility of the protein. According to a previous report, the solubility of egg white proteins is respectively decreased by 4.18%, 4.40%, 7.84%, and 9.66% after treatment with 25 kV/cm HPEF for 200, 400, 600, and 800 ls, respectively [37]. The absolute amount and ratio of hydrophilic and hydrophobic groups determine the hydrophilic or hydrophobic nature of proteins, and the parameters of HPEF affect the functional properties of protein molecular denaturation to some extent.

As shown in Figure 8, 0.25% TGase slightly increased the solubility of PEP (36.42%) compared to that of the control protein (35.76%) (*p* < 0.05), which may be because the binding of protein and TGase can promote the stretching and loosening of the spatial protein structure [55]. However, the solubility of PEP decreased significantly with the increase in TGase addition and reduced to 25.65% after treatment with 4% TGase (*p* < 0.05). A high dose of TGase-catalyzed cross-linking makes PEP form insoluble aggregates or polymers, and the soluble protein content in solution becomes less after being centrifuged [38]. A similar result was obtained in whey protein concentrate–carboxymethylated chitosan composite membranes [56]. Although TGase polymerization results in the formation of specific aggregates that reduces the interfacial dispersion of proteins, the TGase-induced unfolding and extension of the PEP structure significantly increases hydrophobicity owing to promoting the exposure of more hydrophobic groups. The higher surface hydrophobicity of proteins indicates more hydrophobic groups on the protein surface, which tends to reduce protein solubility [57]. Notably, different TGase-induced protein species exhibit different solubility, probably due to the difference in the spatial structure of the substrate proteins.

PEP solubility of the complex treatment (HPEF-1000, 1500, 2000 V/cm; TG-1%) was significantly lower than that of TGase (1%) cross-linking, probably because HPEF treatment (1000, 1500, 2000 V/cm) slightly disrupts and opens the structure of PEP to produce more sites for TGase binding, making the TGase cross-linking reaction more complete. It has been found that the solubility of TGase-cross-linked sWPI decreased slightly, with the solubility of TGase-cross-linked sWPI decreasing by 2.56% compared to TGase-cross-linked WPI at 10 h of ultrafine grinding [9]. This suggests that prolonged ultrafine pulverization time contributes to the production of insoluble aggregates or polymers [9]. In contrast, the solubility of PEP treated with HPEF (2500 V/cm) and TGase cross-linked and TGase cross-linked alone was not significant, suggesting that HPEF may have overly disrupted the structure of PEP, leaving the TGase cross-linking catalysis unimproved.

## 4. Conclusions

In this study, the effects of HPEF, TGase, and HPEF-TGase combination treatment on the solubility, physicochemical, and rheological properties of PEP were investigated for the first time. The results showed that HPEF (500–1500 V/cm) could slightly increase the solubility, free sulfhydryl groups, and particle size of PEP. After TGase cross-linking, a small amount of TGase increased the solubility, free sulfhydryl group, particle size, flow coefficient, and apparent viscosity of PEP. With the increase in TGase addition, the number of larger aggregates, apparent viscosity, elastic modulus, and viscous modulus were increased, while the solubility, free sulfhydryl group, particle size, and loss coefficient decreased continuously. The composite-treated (HPEF-1500 V/cm; 1% TGase) PEP had higher apparent viscosity, elastic modulus, and viscous modulus, and lower solubility, free sulfhydryl group, particle size, and loss coefficient than the TGase cross-linked PEP alone. All the above indicates that HPEF treatment can increase the cross-linking of TGase-induced PEP and improve the rheological properties of PEP by affecting the physicochemical properties, thus making the gel system of PEP solution more stable. This provides a new development idea for the processing of *Pleurotus eryngii* products.

## Figures and Tables

**Figure 1 foods-12-00647-f001:**
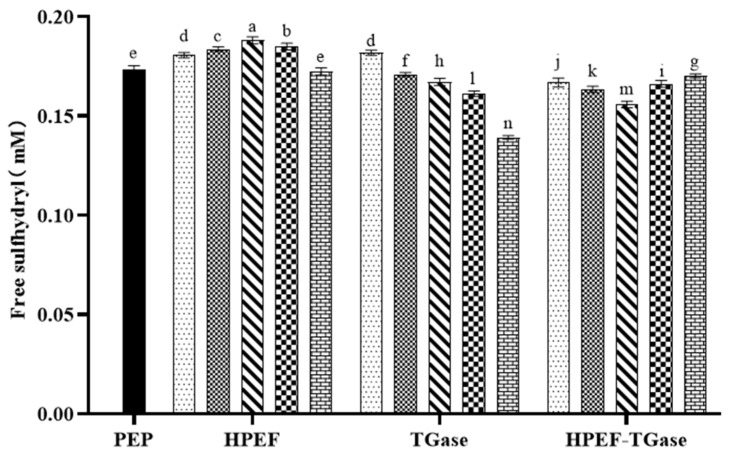
The effects of HPEF (500, 1000, 1500, 2000, and 2500 V/cm) addition, TGase (0.25, 0.5, 1, 2, and 4%), and compound treatment (HPEF-500, 1000, 1500, 2000, 2500 V/cm; 1% TGase) on the free sulfhydryl groups of PEP. 
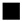
: representative control group PEP. In the first group, 
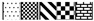
 represent HPEF-PEP of 500, 1000, 1500 2000, and 2500 V/cm, respectively. In the second group, 
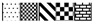
 represent TGase-PEP of 0.25, 0.5, 1, 2, and 4%, respectively. In the third group, 
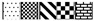
 represent HPEF-TGase-PEP treatments of HPEF-500, 1000, 1500, 2000, and 2500 V/cm, respectively, with 1% TGase. The average particle size group and the solubility are the same as the free sulfhydryl. The different lowercase letters indicate a significant difference between groups (*p* < 0.05).

**Figure 2 foods-12-00647-f002:**
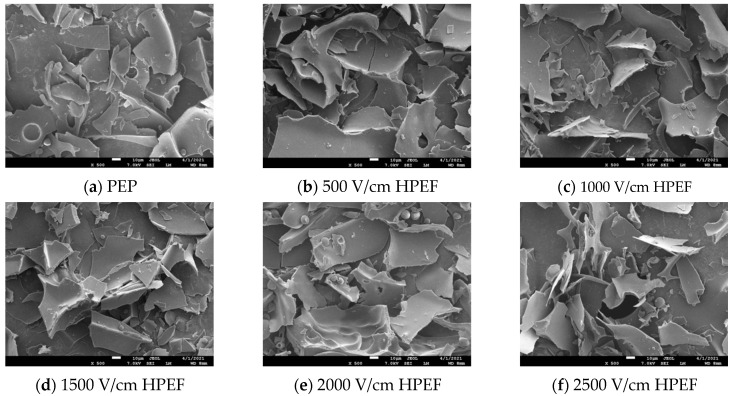
The effects of HPEF (500, 1000, 1500, 2000, and 2500 V/cm) addition (×500), TGase (0.25, 0.5, 1, 2, and 4%) (×1000), and compound treatment (HPEF-500, 1000, 1500, 2000, 2500 V/cm; 1% TGase) (×1000) on the micromorphology of PEP.

**Figure 3 foods-12-00647-f003:**
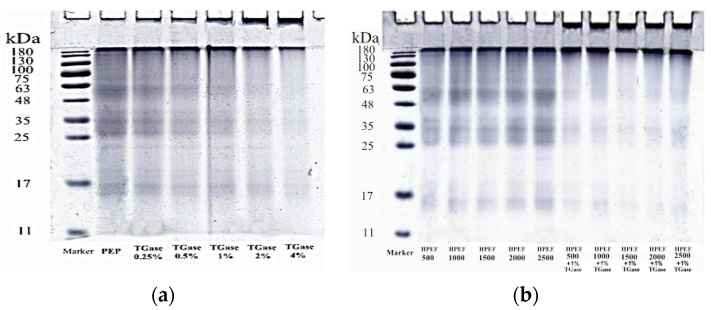
The effects of HPEF (500, 1000, 1500, 2000, and 2500 V/cm) addition, TGase (0.25, 0.5, 1, 2, and 4%), and compound treatment (HPEF-500, 1000, 1500, 2000, 2500 V/cm; 1% TGase) on the molecular weight of PEP. (**a**): The effects of TGase (0.25, 0.5, 1, 2, and 4%); (**b**): The effects of HPEF (500, 1000, 1500, 2000, and 2500 V/cm) addition and compound treatment (HPEF-500, 1000, 1500, 2000, 2500 V/cm; 1% TGase).

**Figure 4 foods-12-00647-f004:**
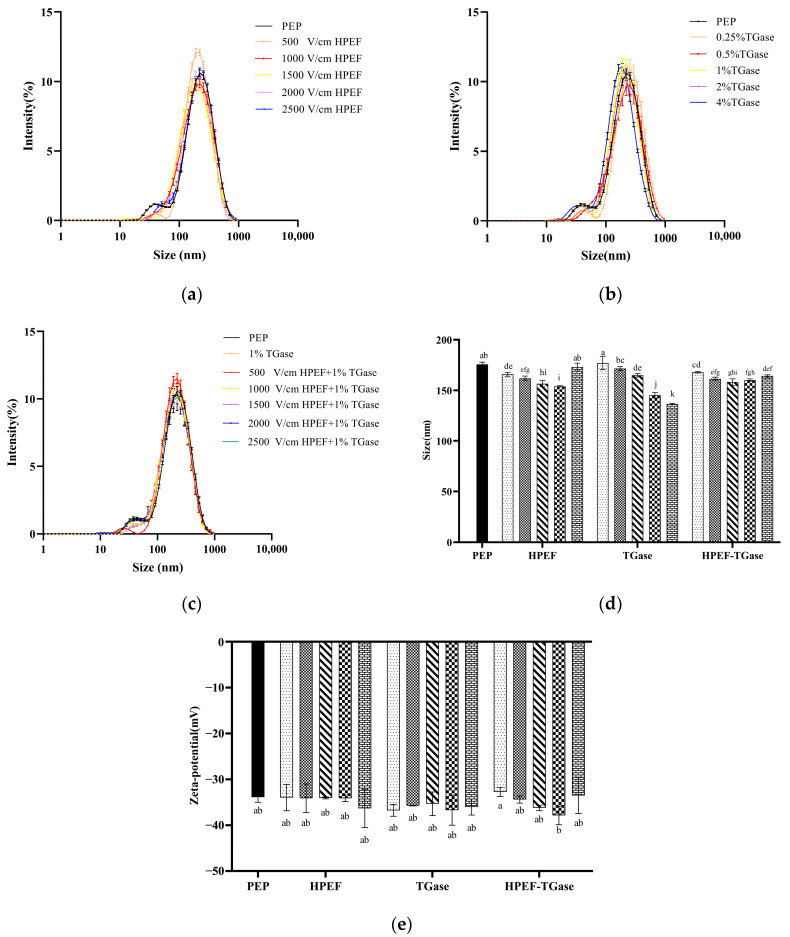
The effects of HPEF (500, 1000, 1500, 2000, and 2500 V/cm) addition, TGase (0.25, 0.5, 1, 2, and 4%), and compound treatment (HPEF-500, 1000, 1500, 2000, 2500 V/cm; 1% TGase) on the particle size distribution, average particle size, and Zeta-potential of PEP. Note: (**a**), particle size distribution curves of PEP after HPEF treatment; (**b**), particle size distribution curves of PEP after TGase treatment; (**c**), particle size distribution curves of PEP after HPEF + TGase treatment; (**d**), average particle size; (**e**), Zeta-potential. The different lowercase letters indicate a significant difference between groups (*p* < 0.05).

**Figure 5 foods-12-00647-f005:**
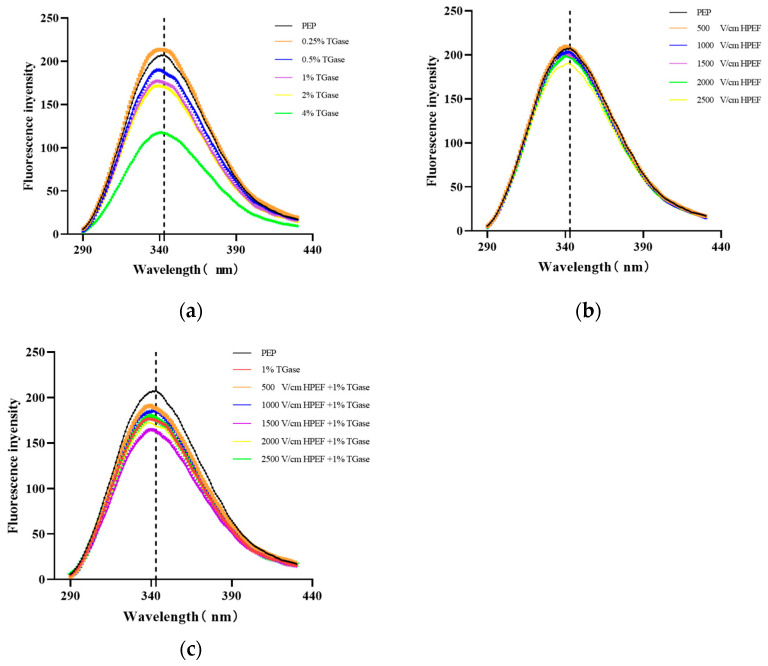
The effects of HPEF (500, 1000, 1500, 2000, and 2500 V/cm) addition, TGase (0.25, 0.5, 1, 2, and 4%), and compound treatment (HPEF-500, 1000, 1500, 2000, 2500 V/cm; 1% TGase) on the fluorescence spectrum of PEP. (**a**): Fluorescence spectrum of PEP treated with HPEF (500, 1000, 1500, 2000, and 2500 V/cm); (**b**): Fluorescence spectrum of PEP treated with TGase (0.25, 0.5, 1, 2, and 4%); (**c**): Fluorescence spectrum of PEP treated with HPEF and TGase (HPEF-500, 1000, 1500, 2000, 2500 V/cm; 1% TGase).

**Figure 6 foods-12-00647-f006:**
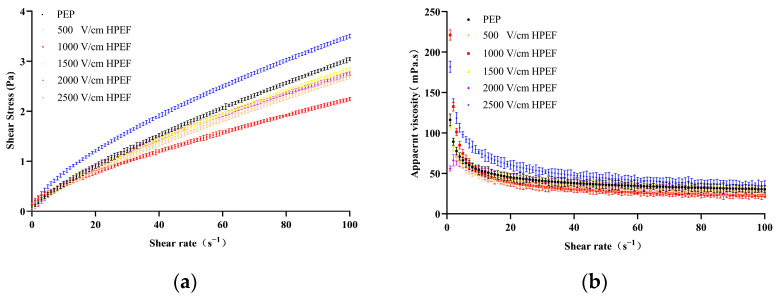
The effects of HPEF (500, 1000, 1500, 2000, and 2500 V/cm), TGase (0.25, 0.5, 1, 2, and 4%), and compound treatment (HPEF-500, 1000, 1500, 2000, 2500 V/cm; 1% TGase) on the shear stress and apparent viscosity of PEP. (**a**): Shear stress of PEP treated with HPEF (500, 1000, 1500, 2000, and 2500 V/cm); (**b**): Apparent viscosity of PEP treated with HPEF (500, 1000, 1500, 2000, and 2500 V/cm)TGase (0.25, 0.5, 1, 2, and 4%); (**c**): Shear stress of PEP treated with TGase (0.25, 0.5, 1, 2, and 4%); (**d**): Apparent viscosity of PEP treated with TGase (0.25, 0.5, 1, 2, and 4%); (**e**): Shear stress of PEP treated with HPEF and TGase (HPEF-500, 1000, 1500, 2000, 2500 V/cm; 1% TGase); (**f**): Apparent viscosity of PEP treated with HPEF and TGase (HPEF-500, 1000, 1500, 2000, 2500 V/cm; 1% TGase).

**Figure 7 foods-12-00647-f007:**
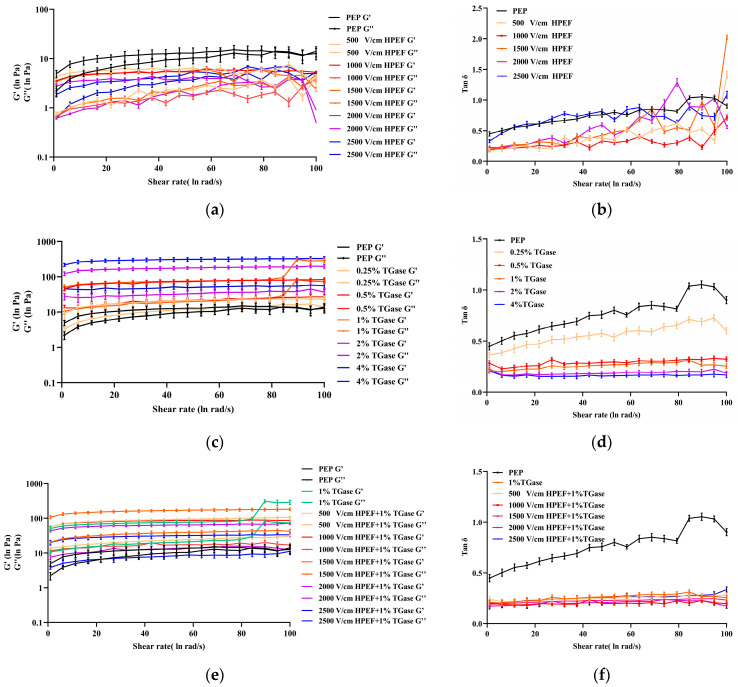
The effects of HPEF (500, 1000, 1500, 2000, and 2500 V/cm), TGase (0.25, 0.5, 1, 2, and 4%), and compound treatment (HPEF-500, 1000, 1500, 2000, 2500 V/cm; 1% TGase) on the G’/G” and tanδ of PEP. (**a**): The G’/G” of PEP treated with HPEF (500, 1000, 1500, 2000, and 2500 V/cm); (**b**): The tanδ of PEP treated with HPEF (500, 1000, 1500, 2000, and 2500 V/cm); (**c**): The G’/G” of PEP treated with TGase (0.25, 0.5, 1, 2, and 4%); (**d**): The tanδ of PEP treated with TGase (0.25, 0.5, 1, 2, and 4%); (**e**): The G’/G” of PEP treated with HPEF and TGase (HPEF-500, 1000, 1500, 2000, 2500 V/cm; 1% TGase); (**f**): The tanδ of PEP treated with HPEF and TGase (HPEF-500, 1000, 1500, 2000, 2500 V/cm; 1% TGase).

**Figure 8 foods-12-00647-f008:**
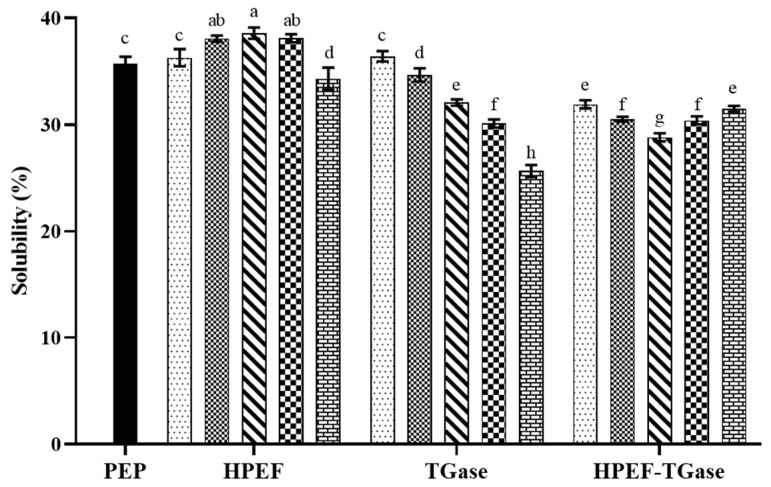
The effects of HPEF (500, 1000, 1500, 2000, and 2500 V/cm), TGase (0.25, 0.5, 1, 2, and 4%), and compound treatment (HPEF-500, 1000, 1500, 2000, 2500 V/cm; 1% TGase) on the solubility of PEP. The different lowercase letters indicate a significant difference between groups (*p* < 0.05).

**Table 1 foods-12-00647-t001:** The effect of HPEF (500, 1000, 1500, 2000, and 2500 V/cm) addition, TGase (0.25, 0.5, 1, 2, and 4%), and compound treatment (HPEF-500, 1000, 1500, 2000, 2500 V/cm; 1% TGase) on the secondary structure of PEP.

PEP	Secondary Structure Content/%		
	α-Helix	β-Fold	β-Turn
PEP	26.6 ± 0.2	38.1 ± 0.3	35.3 ± 0.5
500 V/cm HPEF	28.2 ± 0.6 *	35.6 ± 1.0 *	36.2 ± 0.2
1000 V/cm HPEF	28.4 ± 0.6 *	35.4 ± 0.5 *	36.1 ± 0.2
1500 V/cm HPEF	27.6 ± 0.8 *	35.5 ± 0.7 *	37.0 ± 0.6 *
2000 V/cm HPEF	27.2 ± 0.6	35.8 ± 0.4 *	37.0 ± 0.8 *
2500 V/cm HPEF	26.2 ± 0.3	36.5 ± 0.5 *	37.4 ± 0.6 *
0.25% TG	29.6 ± 0.5 *	30.3 ± 0.6 *	40.2 ± 0.3 *
0.5% TG	30.7 ± 1.0 *	31.6 ± 0.4 *	37.7 ± 0.2 *
1% TG	29.3 ± 0.4 *	31.8 ± 0.2 *	38.9 ± 1.1 *
2% TG	29.9 ± 0.7 *	32.6 ± 0.5 *	37.5 ± 0.6 *
4% TG	30.3 ± 0.6 *	32.2 ± 0.4 *	36.6 ± 0.5 *
500 HPEF + 1% TGase	29.1 ± 0.4	32.7 ± 0.2 ^#^	38.0 ± 0.5
1000 HPEF + 1% TGase	28.7 ± 0.2	31.5 ± 0.4	39.8 ± 0.4 ^#^
1500 HPEF + 1% TGase	29.3 ± 0.4	32.3 ± 0.3	38.5 ± 0.9
2000 HPEF + 1% TGase	29.5 ± 0.6	32.5 ± 0.5	38.0 ± 0.7
2500 HPEF + 1% TGase	29.0 ± 0.5	33.1 ± 0.5 ^#^	37.9 ± 0.1

* *p* < 0.05, compared with PEP; ^#^
*p* < 0.05, compared with 1% TG.

**Table 2 foods-12-00647-t002:** Power-law model of PEP under different HPEF (500, 1000, 1500, 2000, and 2500 V/cm) pretreatments.

HPEF (V/cm)	K	N	R^2^
PEP	0.0885	0.7659	0.99999
500 HPEF	0.0737	0.7778	0.99991
1000 HPEF	0.0636	0.7582	0.99255
1500 HPEF	0.0904	0.7469	0.99992
2000 HPEF	0.1284	0.6710	0.99977
2500 HPEF	0.1593	0.6698	0.99998

Note: K: Viscosity coefficient; N: Flow behavior index.

**Table 3 foods-12-00647-t003:** Power-law model of PEP with different TGase content (0.25, 0.5, 1, 2, 4%).

TGase (%)	K	N	R^2^
PEP	0.0885	0.7659	0.99999
0.25	0.1318	0.7819	0.99964
0.5	0.2389	0.6799	0.99861
1	0.5345	0.5931	0.99971
2	0.7346	0.6626	0.99992
4	0.9309	0.6522	0.99976

Note: K: Viscosity coefficient; N: Flow behavior index.

**Table 4 foods-12-00647-t004:** The power-law model of PEP under the preconditioning of compound treatment (HPEF-500, 1000, 1500, 2000, 2500 V/cm; 1% TGase).

TGase-HPEF	K	N	R^2^
PEP	0.0885	0.7659	0.99999
1% TGase	0.5345	0.5931	0.99971
500 HPEF + 1% TGase	0.5347	0.6193	0.99999
1000 HPEF + 1% TGase	0.6141	0.6032	0.99899
1500 HPEF + 1% TGase	0.8546	0.6789	0.99956
2000 HPEF + 1% TGase	0.4098	0.5540	0.99657
2500 HPEF + 1% TGase	0.3004	0.6819	0.99961

Note: K: Viscosity coefficient; N: Flow behavior index.

## Data Availability

All original data included in this paper are available from the authors upon reasonable request.

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
