# Peer review of "Investigation of Consequences of High-Voltage Pulsed Electric Field and TGase Cross-Linking on the Physicochemical and Rheological Properties of Pleurotus eryngii Protein"

_foods, 2023, doi:10.3390/foods12030647_

Round 1

Reviewer 1 Report

Dear Authors, 

The Manuscript entitled "Investigation of consequences of high-voltage pulsed electric field and TGase cross-linking on the physicochemical and rheological properties of Pleurotus eryngii protein" presents an interesting approach on green technologies, Tgase and protein function/stracture. However, there are several points need to be clarified. In general: 

- The overall quality of the pictures is very low. Please fix this. 
- In most of the paragraph, section "Results and Discussion", you need to include more comparison with the literature. At this stage, it's only a presentation of your results. The reference list is OK however, please add some value from the previous published papers. 

In specific:
- Line 72. I don't think that you explain the theretical basis in this Manuscript. Pelase, explain this sentence. 
- Paragraph 2.2. pH values? Time? 
- Paragraph 3.2, lines 224-225. Please, explain this sentence. In addition, based on the figures the only difference is between PEF treatment and Tgase. Could you please explain this?
- Particle Size Distribution. Please, provide the PDI values for all the measures. Then, discuss based on this. 
- Figure 4. Please, fix the number (a), (b).... in the figure caption. 
- Paragraph 3.5. Reference with the literature?
- Paragraph 3.7. Discussion with the literature? In addtion, there are a lot of "increase" and "decrease" in this section. Pelase, change it. 

Thanks a lot. 

Best, 

Author Response

Response to Reviewer 1 Comments

  Thank you for your letter and for the reviewer’s comments concerning our manuscript entitled “Investigation of consequences of high-voltage pulsed electric field and TGase cross-linking on the physicochemical and rheological properties of Pleurotus eryngii protein” (foods-2098634). Those comments are all valuable and very helpful for revising and improving our paper. We have studied reviewers’ comments carefully and have tried our best to revise our manuscript according to the comments point-by-point, and discussed the subsequent modification. In the following, the original comments are presented in italic font and our responses are in regular roman font. Modifications are put in a separate paragraph started with “Revisions” which are given in bold font. All revisions to the manuscript were marked up using the “Track Changes” function. The main corrections in the manuscript and the responses to the editor’s comments are as follows:

The Manuscript entitled "Investigation of consequences of high-voltage pulsed electric field and TGase cross-linking on the physicochemical and rheological properties of Pleurotus eryngii protein" presents an interesting approach on green technologies, Tgase and protein function/stracture. However, there are several points need to be clarified. In general: 

Response: We appreciate the acknowledgement from this review on the merit of our work. We have tried our best to address every concern from this reviewer.

- The overall quality of the pictures is very low. Please fix this. 

Response: Thanks for your suggestion. We have tried our best to fix the quality of the pictures.

Figure 1. The effects of HPEF (500, 1000, 1500, 2000 and 2500 V/cm) addition, TGase (0.25, 0.5, 1, 2 and 4%) and compound treatment (HPEF-500, 1000, 1500, 2000, 2500 V/cm;TGase-1%) on the free sulfhydryl groups of PEP.representative control group PEP;The first group  represents (HPEF-PEP: 500, 1000, 1500 2000 and 2500V/cm)ï¼›The second group  represents (TGase-PEP: 0.25, 0.5, 1, 2 and 4%);The third group  represents (HPEF-TGase-PEP: HPEF-500, 1000, 1500, 2000, 2500 V/cm;TGase-1%). The average particle size group and the solubility are the same as the free sulfhydryl.

  • PEP                (b)500 V/cm HPEF             (c)1000 V/cmHPEF

(d)1500 V/cm HPEF          (e)2000 V/cm HPEF          (f)2500 V/cm HPEF

(g)PEP              ï¼ˆ(h)0.25% TGase            ï¼ˆ(i)0.5%TGase

(j)1% TGase                  (k)2% TGase                 (l)4% TGase

(m)1% TGase         (n)500 V/cm HPEF+1%TGase  (o)1000 V/cm HPEF+1%TGase

(p)1500 V/cm HPEF+1% TGase  (q)2000 V/cm HPEF+1%TGase  (r)2500 V/cm HPEF+1%TGase

Figure 2. The effects of HPEF (500, 1000, 1500, 2000 and 2500 V/cm) addition (×500), TGase (0.25, 0.5, 1, 2 and 4%) (×1000) and compound treatment (HPEF-500, 1000, 1500, 2000, 2500 V/cm;TGase-1%) (×1000) on the micromorphology of PEP.

Figure 3. The effect of HPEF (500,1000,1500, 2000 and 2500 V/cm) addition, TGase (0.25,0.5,1, 2 and 4%) and compound treatment (HPEF-500,1000,1500,2000,2500 V/cm;TGase-1%) on the molecular weight of PEP.

(a)

(b)

(c)

                                          (d)

                                              (e)

Figure 4. The effects of HPEF (500, 1000, 1500, 2000 and 2500 V/cm) addition, TGase (0.25, 0.5, 1, 2 and 4%) and compound treatment (HPEF-500, 1000, 1500, 2000, 2500 V/cm; TGase-1%) on the particle size distribution, average particle size, and Zeta-potential of PEP.

Note: a, particle size distribution curves of PEP after HPEF treatment; b, particle size distribution curves of PEP after TGase treatment; c, particle size distribution curves of PEP after HPEF+TGase treatment; d, average particle size; e, Zeta-potential.

                            (a)

                             (b)

                              (c)

Figure 5. The effects of HPEF (500, 1000, 1500, 2000 and 2500 V/cm) addition, TGase (0.25, 0.5, 1, 2 and 4%) and compound treatment (HPEF-500, 1000, 1500, 2000, 2500 V/cm; TGase-1%) on the fluorescence spectrum of PEP.

(a)

(b)

(c)

(d)

(e)

(f)

Figure 6. The effects of HPEF (500, 1000, 1500, 2000 and 2500 V/cm), TGase (0.25, 0.5, 1, 2 and 4%) and compound treatment (HPEF-500, 1000, 1500, 2000, 2500 V/cm; TGase-1%) on the shear stress and apparent viscosity of PEP.

(a)

(b)

(c)

(d)

(e)

(f)

Figure 7. The effects of HPEF (500, 1000, 1500, 2000 and 2500 V/cm), TGase (0.25, 0.5, 1, 2 and 4%) and compound treatment (HPEF-500, 1000, 1500, 2000, 2500 V/cm; TGase-1%) on the G'/G " and tanδ of PEP.

Figure 8. The effects of HPEF (500, 1000, 1500, 2000 and 2500 V/cm), TGase (0.25, 0.5, 1, 2 and 4%) and compound treatment (HPEF-500, 1000, 1500, 2000, 2500 V/cm; TGase-1%) on the solubility of PEP.

- In most of the paragraph, section "Results and Discussion", you need to include more comparison with the literature. At this stage, it's only a presentation of your results. The reference list is OK however, please add some value from the previous published papers. 

Response: Thanks for your suggestion. We have added the corresponding discussion and references.

Revisions:

  • Line 231-232: We added the sentence of Similar results were obtained in the study of whey-soybean mixed protein [29]. to supplement the sentence of In addition, the increase of cross-linked aggregates may correlate with the decrease of PEP solubility after TGase cross-linking.
  • Line 256: We added the sentence of “coincident with Qian [18] who investigated the PEF on egg white protein.”at the end of the paragraph.
  • Line 288-289: The same results were obtained in the study of soybean protein isolates, which may be due to the dissociation and reaggregation of proteins induced by PEF treatment[35].at the end of the paragraph.
  • Line 344-346: We added the sentence of “Similarly, Zhang et al [32] found that the TGase cross-linking reaction had no significant impact on the zeta-potential of whey protein soluble aggregates.”at the end of the paragraph.
  • Line 366-368: We added the sentence of “It has also been suggested that PEF treatment will expose more previously buried tryptophan residues[18].”at the middle of the paragraph.
  • In the part of References: We added the reference of Han, X.; Liang, Z.Q.; Tian, S.F.; Liu, L.; Wang, S. Modification of whey-soybean mixed protein by sequential high-pressure homogenization and transglutaminase treatment. LWT 2022, 172, 114217. https://doi.org/10.1016/j.lwt.2022.114217.
  • In the part of References: We added the reference of Akbari, A.; Wu, J. Cruciferin nanoparticles: Preparation, characterization and their potential application in delivery of bioactive compounds. Food Hydrocoll. 2016, 54, 107-118. https://doi.org/10.1016/j.foodhyd.2015.09.017.

Comment 1. Line 72. I don't think that you explain the theoretical basis in this Manuscript. Please, explain this sentence.

Response 1: Thanks for your suggestion. Since the reviewer think that we don’t explain the theoretical basis in this manuscript, we rewrite the sentence.

Revision:

(1)Line 72-73: The sentence was changed to “and provide a reference for improving the comprehensive utilization of Pleurotus eryngii.”.

Comment 2. Paragraph 2.2. pH values? Time?

Response 2: Thanks for your suggestion. We added the extraction time of PEP and the isoelectric points of albumin, globulin and gluten.

Revisions:

  • Line 86: We added “2.5 h, 2 h and 3 h orderly”after “and alkaline solution for”.
  • Line 87: We added “at isoelectric points 3.9, 4.2 and 4.3 respectively”after “albumin, globulin, and glutenin were obtained”.

Comment 3. Paragraph 3.2, lines 224-225. Please, explain this sentence. In addition, based on the figures the only difference is between PEF treatment and TGase. Could you please explain this?

Response 3: â‘ As suggested by the reviewer, we added the reference to explain the sentence of “In addition, the increase of cross-linked aggregates may correlate with the decrease of PEP solubility after TGase cross-linking.”, referenced as [29].

â‘¡It can be intuitively observed from the figure that compared with the untreated PEP, the PEP is still in a lamellar structure after HPEF treatment, but the dense lamellar sheet begins to thicken and become loose, with more irregular irregular shapes and more fragments. After TGase cross-linking, the protein formed a dense and uniform honeycomb structure. This is mainly because HPEF treatment will expand PEP molecules and make them more fluffy; TGase catalyzes the formation of covalent cross-linking between protein molecules to form intramolecular or intermolecular network structure.

Revisions:

  • Line 233-234: We added the sentence of Similar results were obtained in the study of whey-soybean mixed protein [29]. to supplement the sentence of In addition, the increase of cross-linked aggregates may correlate with the decrease of PEP solubility after TGase cross-linking.
  • Line 220-221: We changed the sentences of These changes may be owing to the electric field making it fluffy, with the most significant change at the HPEF intensity of 2500 V/cm (Figure 2a-2f). to These changes may be owing to the electric field expending it molecules and make them more fluffy, with the most significant change at the HPEF intensity of 2500 V/cm (Figure 2a-2f). .
  • Line 224-226: We changed the sentences of TGase catalyzes the formation of covalent cross-linking between protein molecules to form intramolecular or intermolecular network structure.after the sentences of The extracted and purified PEP showed a thin sheet structure (Figure 2g), while the PEP showed a network structure after TGase cross-linking (Figure 2h-2l).
  • In the part of References: We added the reference of Han, X.; Liang, Z.Q.; Tian, S.F.; Liu, L.; Wang, S. Modification of whey-soybean mixed protein by sequential high-pressure homogenization and transglutaminase treatment. LWT 2022, 172, 114217. https://doi.org/10.1016/j.lwt.2022.114217.

Comment 4. Particle Size Distribution. Please, provide the PDI values for all the measures. Then, discuss based on this.

Response 4: Thanks for your suggestion. We used the particle size distribution and average particle size to represent the particle size for all the measures. I am sorry for not determining the PDI values. We will pay attention to this problem in our future study.

Comment 5. Figure 4. Please, fix the number (a), (b).... in the figure caption.

Response 5: Thanks for your careful checks. We are sorry for our carelessness. In the revised manuscript, we have fixed the position of the serial number to ensure that it corresponds to the figure position.

Revisions:

  • In Figure 4: We added the sentences of “Note: a, particle size distribution curves of PEP after HPEF treatment; b, particle size distribution curves of PEP after TGase treatment; c, particle size distribution curves of PEP after HPEF+TGase treatment; d, average particle size; e, Zeta-potential. ”.

Comment 6. Paragraph 3.5. Reference with the literature?

Response 6: We sincerely appreciate the valuable comments. We have checked the literature carefully and added the 34th reference for the sentences of “The value of the Zeta potential can be significantly correlated with the stability of a colloidal dispersion and indicates the degree of repulsion between neighboring substances and similarly charged particles” and added the zeta-potential of whey protein soluble aggregates to support this view.

Revisions:

  • Line 339: We added[36] for the sentences of “The value of the Zeta potential can be significantly correlated with the stability of a colloidal dispersion and indicates the degree of repulsion between neighboring substances and similarly charged particles”.
  • Line 347-349: We added the sentence of “Similarly, Zhang et al [33] found that the TGase cross-linking reaction had no significant impact on the zeta-potential of whey protein soluble aggregates.”at the end of the paragraph.
  • In the part of References: We added the reference of Akbari, A.; Wu, J. Cruciferin nanoparticles: Preparation, characterization and their potential application in delivery of bioactive compounds. Food Hydrocoll. 2016, 54, 107-118. https://doi.org/10.1016/j.foodhyd.2015.09.017.

Comment 7. Paragraph 3.7. Discussion with the literature? In addition, there are a lot of "increase" and "decrease" in this section. Please, change it.

Response 7: â‘ As suggested by the reviewer, we added more references to explain the idea, referenced as [42] and [43].

â‘¡We sincerely thank the reviewer for careful reading. As suggested by the reviewer, we have changed the "increase" to "growth", "enhance", "add" "raise" and "decrease" to "decline", "reduce".

Revisions:

  • Line 406: We added the reference of “[22,42]”for the sentence of “the changes in the secondary structure of PEP before and after treatment were investigated by analyzing the planar bending of NH and the change of -CN length in the amide one band (1700-1600 cm-1) region of the IR spectrum and the change of C=O length in the structure.”.
  • Line 408-413: We changed the sentences of “As shown in Table 1, HPEF increased α-helix and β-turn but decreased β-fold of PEP. With the increase of the HPEF strength, the α-helix and β-fold gradually decreased, and the β-turn continuously increased. Moreover, the overall decrease of the β-fold was equal to the increase of α-helix and β-turn, which indicates that HPEF might make the PEP structure less rigid and more flexible. Similar results were obtained in a previous study of rapeseed globulin [13].”to “As shown in Table 1, HPEF increased α-helix and β-turn but decreased β-fold of PEP. With the growth of the HPEF strength, the α-helix and β-fold gradually reduced, and the β-turn continuously enhanced. Moreover, the overall decline of the β-fold was equal to the add of α-helix and β-turn, which indicates that HPEF might make the PEP structure less rigid and more flexible. Similar results were obtained in a previous study of rapeseed globulin [13].”.
  • Line 423: We added the reference of [43]for the sentence of “The presence of more hydrogen bonds in α-helix and β-fold makes the protein structure exhibit a certain rigidity, while the β-turn and irregular curl show great flexibility because they do not contain hydrogen bonds and other interactions.”.
  • Line 418-429: We changes the sentence of “After PEP was cross-linked by TGase, the α-helix and β-turn increased, and the β-fold decreased, which indicates that TGase cross-linking may promote the transformation of β-fold to α-helix and β-turn. The presence of more hydrogen bonds in α-helix and β-fold makes the protein structure exhibit a certain rigidity, while the β-turn and irregular curl show great flexibility because they do not contain hydrogen bonds and other interactions. A study by Marcoa and Rosell [39] also reported that large polymers and aggregates formed by extensive TGase cross-linking reactions might reduce the flexibility of the protein. The change in the secondary structure content of PEP after modification shows that the increase in α-helix is significantly less than the decrease in β-fold, so TGase may make the PEP structure less rigid and more flexible. However, TGase cross-linking induced a decrease in α-helix and β-turn and an increase in β-fold compared to untreated PPI ”to “After PEP was cross-linked by TGase, the α-helix and β-turn enhanced, and the β-fold reduced, which indicates that TGase cross-linking may promote the transformation of β-fold to α-helix and β-turn. The presence of more hydrogen bonds in α-helix and β-fold makes the protein structure exhibit a certain rigidity, while the β-turn and irregular curl show great flexibility because they do not contain hydrogen bonds and other interactions [41]. A study by Marcoa and Rosell [42] also reported that large polymers and aggregates formed by extensive TGase cross-linking reactions might reduce the flexibility of the protein. The change in the secondary structure content of PEP after modification shows that the increase in α-helix is significantly less than the decrease in β-fold, so TGase may make the PEP structure less rigid and more flexible. However, TGase cross-linking induced a decline in α-helix and β-turn and a raise in β-fold compared to untreated PPI”.
  • In the part of References: We added the reference of Wang, H.; You, S.P.; Wang, W.H.; Zeng, Y.; Su, R.X.; Qi, W.; Wang, K.; He, Z.M. Laccase-catalyzed soy protein and gallic acid complexation: Effects on conformational structures and antioxidant activity. Food Chem. 2022, 375, 131865. https://doi.org/10.1016/j.foodchem.2021.131865.
  • In the part of References: We added the reference of Vega-Mercado, H.; Powers, J.; Barbosa-Cánovas, G.V.; Swanson, B.G,; Luedecke, L. Inactivation of a protease from Pseudomonas fluorescens M3/6 using high voltage pulsed electric fields. In Proceedings IFT Annual Meeting, California, USA , 1995. .

We tried our best to improve the manuscript and made some changes in the manuscript. We appreciate for Editor/Reviewer’s warm work earnestly, and hope the correction will meet with approval. Once again, thank you very much for your comments and suggestions.

Yours sincerely,

Cui-ping Feng

Reviewer 2 Report

The current manuscript reports the investigation of consequences of high-voltage pulsed electric field and TGase cross-linking on the physicochemical and rheological properties of Pleurotus eryngii protein.

In general, this is an important and interesting work, well-written, logically structured. I have, however a few comments or suggestions.

1.     In the last paragraph of the Introduction, state the purpose of the study.

2.     Line 67 - is a typo

3.     Subsection 2.4 indicate the brands and manufacturers of the equipment used

4.     Line 151 - indicate the number of samples to be tested in grams or milliliters

5.     Figure 2 shows the 1% TGase sample twice. Can divide this drawing into 2 parts with different magnifications and be sure to indicate the magnification in the figure caption.

Author Response

Response to Reviewer 2 Comments

  Thank you for your letter and for the reviewer’s comments concerning our manuscript entitled “Investigation of consequences of high-voltage pulsed electric field and TGase cross-linking on the physicochemical and rheological properties of Pleurotus eryngii protein” (foods-2098634). Those comments are all valuable and very helpful for revising and improving our paper. We have studied reviewers’ comments carefully and have tried our best to revise our manuscript according to the comments point-by-point, and discussed the subsequent modification. In the following, the original comments are presented in italic font and our responses are in regular roman font. Modifications are put in a separate paragraph started with “Revisions” which are given in bold font. All revisions to the manuscript were marked up using the “Track Changes” function. The main corrections in the manuscript and the responses to the editor’s comments are as follows:

The current manuscript reports the investigation of consequences of high-voltage pulsed electric field and TGase cross-linking on the physicochemical and rheological properties of Pleurotus eryngii protein.

In general, this is an important and interesting work, well-written, logically structured. I have, however a few comments or suggestions.

Response: We appreciate the acknowledgement from this review on the merit of our work. We have tried our best to address every concern from this reviewer.

Comment 1. In the last paragraph of the Introduction, state the purpose of the study.

Response 1: Thanks for your suggestion. We have stated the purpose of the study in the last paragraph of the introduction.

Revisions:

  • Line 66: We added “to better improve the structure and functional properties of PEP, ”before “complex modification of PEP and TGase was used to explore the effects of HPEF and TGase on the relationship between microstructure and functional properties of PEP treated with HPEF and TGase, ”.

Comment 2. Line 67 - is a typo

Response 2: We were really sorry for our careless mistakes. In our revised manuscript, the typo is revised. Thanks for your correction.

Comment 3. Subsection 2.4 indicate the brands and manufacturers of the equipment used.

Response 3: We are very grateful to this reviewer for reviewing the paper so carefully. We have added the brands and manufacturers of the equipment used.

Revisions:

  • Line 123: We added “(JEOL Electronics Co., Ltd, Japan)”after “a JSM-6490LV SEM”.
  • Line 131: We added “(Mini-PROTEAN, Bio-rad Co., USA)”after “the electrophoresis bath”.
  • Line 148-149: We added “(Madison Nicolet Is 10, Thermo Nicolet Co., USA)”after “the FTIR spectrometer”.

Comment 4. Line 151 - indicate the number of samples to be tested in grams or milliliters.

Response 4: Thanks for your suggestion. According to your suggestion, "Eight percent of protein samples were determined" has been replaced with "Samples were obtained by dissolving 8 g of PEP powder in 100 mL of distilled water".

Revisions:

  • Line 155-156: The sentence was changed to “Samples were obtained by dissolving 8 g of PEP powder in 100 mL of distilled water”.

Comment 5. Figure 2 shows the 1% TGase sample twice. Can divide this drawing into 2 parts with different magnifications and be sure to indicate the magnification in the figure caption.

Response 5: Thanks for your careful checks. Because we have three different treatment in this study, including HPEF (500, 1000, 1500, 2000 and 2500 V/cm) addition (×500), TGase (0.25, 0.5, 1, 2 and 4%) (×1000) and compound treatment with different HPEF and 1% TGase (HPEF-500, 1000, 1500, 2000, 2500 V/cm;TGase-1%) (×1000), thus we showed the 1% TGase sample twice. We have replaced one of the 1% TGase images. Moreover, the images of PEP treated with HPEF can only be seen clearly at the magnification of ×500, but the images of PEP treated with 1% TGase cann’t be seen clearly at the magnification of ×500. The effects of PEP treated with HPEF and 1% TGase or 1% TGase was the best at the magnification of ×1000. In addition, the HPEF is an auxiliary treatment to promote TGase cross-linking. We concentrate on the compare of the effects between TGase and composite treatment. So the The effects of HPEF were observed at the magnification of ×500 and the effects of TGase or compound treatment were observed at the magnification of ×1000. Besides, we added the magnification in the figure caption.

Revisions:

  • Figure 2: The second image of 1% TGase was replaced.

(m)1% TGase 

(2)Figure 2: The magnification in the figure caption was added. The effects of HPEF (500, 1000, 1500, 2000 and 2500 V/cm) addition (×500), TGase (0.25, 0.5, 1, 2 and 4%) (×1000) and compound treatment (HPEF-500, 1000, 1500, 2000, 2500 V/cm;TGase-1%) (×1000) on the micromorphology of PEP.

  We tried our best to improve the manuscript and made some changes in the manuscript. We appreciate for Editor/Reviewer’s warm work earnestly, and hope the correction will meet with approval. Once again, thank you very much for your comments and suggestions.

Yours sincerely,

Cui-ping Feng
